# Achieving High Activity and Long-Term Stability towards Oxygen Evolution in Acid by Phase Coupling between CeO_2_-Ir

**DOI:** 10.3390/ma16217000

**Published:** 2023-11-01

**Authors:** Jianren Kuang, Zhi Li, Weiqiang Li, Changdong Chen, Ming La, Yajuan Hao

**Affiliations:** 1College of Environment and Energy, South China University of Technology, Guangzhou 510006, China; kjr_be_a_man1998@163.com (J.K.); m19927657431@163.com (Z.L.); 2College of Electric and Information Engineering, Pingdingshan University, Pingdingshan 467000, China; 5001@pdsu.edu.cn; 3College of Chemistry and Environmental Engineering, Pingdingshan University, Pingdingshan 467000, China; lmccd5613@163.com

**Keywords:** CeO_2_-Ir heterojunction, electrocatalysis, acidic oxygen evolution reaction, high activity and long-term stability

## Abstract

The development of efficient and stable catalysts with high mass activity is crucial for acidic oxygen evolution reaction (OER). In this study, CeO_2_-Ir heterojunctions supported on carbon nanotubes (CeO_2_-Ir/CNTs) are synthesized using a solvothermal method based on the heterostructure strategy. CeO_2_-Ir/CNTs demonstrate remarkable effectiveness as catalysts for acidic OER, achieving 10.0 mA cm^−2^ at a low overpotential of only 262.9 mV and maintaining stability over 60.0 h. Notably, despite using an Ir dosage 15.3 times lower than that of c-IrO_2_, CeO_2_-Ir/CNTs exhibit a very high mass activity (2542.3 A g_Ir_^−1^@1.53 V), which is 58.8 times higher than that of c-IrO_2_. When applied to acidic water electrolyzes, CeO_2_-Ir/CNTs display a prosperous potential for application as anodic catalysts. X-ray photoelectron spectrometer (XPS) analysis reveals that the chemical environment of Ir nanoparticles (NP) can be effectively modulated through coupling with CeO_2_. This modulation is believed to be the key factor contributing to the excellent OER catalytic activity and stability observed in CeO_2_-Ir/CNTs.

## 1. Introduction

A significant amount of renewable energy (such as wind energy, tidal energy, solar energy, etc.) is wasted due to ineffective utilization. Utilizing energy storage and conversion devices to convert renewable energy into other forms of stored energy represents an effective approach for addressing the aforementioned issues [1,2]. A proton exchange membrane water electrolyzer (PEMWE) offers the capability to convert renewable energy into chemical energy in hydrogen, which possesses sustainable, clean, and efficient characteristics [3,4]. PEMWE comprises two crucial half-reactions: cathodic hydrogen evolution reaction (HER) and anodic oxygen evolution reaction (OER). OER involves a four electron–proton transfer process that includes complex intermediates’ conversions and necessitates high kinetic costs [5,6]. Most catalysts reported thus far encounter a common challenge: low catalytic activity and poor stability in acidic OER [7,8]. In view of the comprehensive consideration of catalytic activity and stability, Ir-based materials are widely used as catalysts in acidic OER. However, these catalysts currently face challenges such as scarcity, high cost, insufficient catalytic activity, and poor stability, hence they still have a long way to go before practical application can be achieved [9,10]. Consequently, exploring low cost Ir-based catalysts with high activity and durability has become a challenge at present.

It has been reported that metallic Ir catalysts with nanorods [11], nanosheets [12], and other structures exhibit exceptional catalytic activity and stability in acidic OER [13]. Significant efforts have been devoted to minimizing the use dosage of Ir without compromising the catalytic activity. Downsizing Ir to the nanometer scale can enhance the exposure of active sites, thereby improving catalytic activity [14]. Furthermore, the construction of heterojunctions could induce lattice strain within the active phase and allow for the regulation of charge distribution between the two phases, resulting in a regulation of their catalytic activity [15,16]. Consequently, the electronic structure of Ir can be modulated through the construction of heterojunctions with other phases, indirectly influencing the adsorption and desorption of intermediates [16,17]. Appropriate adsorption/desorption facilitates conversion processes of intermediates, leading to enhanced reaction rates and accelerated kinetics [18,19]. The construction of a heterojunction between Ir and metal oxides is a common strategy for enhancing both catalytic activity and stability in Ir-based catalysts. Lee et al. achieved improved OER performance by combining Ir with MoO_3_ to construct highly electron-deficient Ir NPs [16]. Xing and colleagues uniformly deposited an ultra-fine (~1.0 nm) layer of Ir metal onto the surface of niobium oxide (Nb_2_O_5−x_), which exhibited a high concentration of oxygen vacancies [20]. During OER, dynamic migration of oxygen takes place at the two-phase interface between Ir and Nb_2_O_5−x_, effectively preventing excessive oxidation of Ir to high-valent species and subsequent deactivation. Although extensive research has focused on developing heterogeneous catalysts based on transition metal oxides combined with iridium (Ir/MO_x_), investigations into heterojunctions formed by rare earth metal oxides and iridium remain scarce. Cerium oxide is widely used as a catalyst and cocatalyst for many oxidation reactions due to its excellent redox properties [21,22,23,24]. For example, Im and colleagues improved the electrocatalytic activity of OER by inducing site-selective crystal disorder through the doping of Ce^3+^ ions into the MIL-88B(Ni) framework [25]. The aforementioned statements have prompted us to redirect our research focus towards CeO_2_ in order to investigate the electron transfer between CeO_2_ and Ir, as well as its impact on both the activity and the stability of OER. Moreover, carbon materials are widely acknowledged as ideal supports for electrocatalysis owing to their remarkable specific surface area and excellent electrical conductivity [26,27,28]. As an exemplary carrier, CNTs offer abundant nucleation sites and conductive pathways for catalytic active materials, thereby significantly enhancing the catalytic activity of catalysts while reducing their required quantity [29,30].

In this study, we propose a strategy for the construction of heterojunctions and carbon combination to enhance catalytic performance while reducing the consumption of Ir. This strategy involves modulating the electronic structure of Ir and improving the electron conduction of the catalyst. To achieve this, we engineer electron-deficient Ir NPs through the construction of heterojunctions with CeO_2_, complemented by the incorporation of CNTs to facilitate efficient charge conduction and improve the utilization of Ir. As a result, CeO_2_-Ir/CNTs exhibit impressive durability and catalytic activity, operating continuously at 10.0 mA cm^−2^ for over 60.0 h with only 262.9 mV overpotential. These findings underscore the excellent catalytic performance and long-term robustness of CeO_2_-Ir/CNTs when employed in acidic water electrolyzers.

## 2. Materials and Methods

### 2.1. Chemicals

Multi-walled carbon nanotubes (CNTs, >97.0%) were bought from Shenzhen Nanotech Port Co., LTD, Shenzhen, China. Iridium (III) chloride (IrCl_3_, with Ir ≥ 62.0%) and cerous chloride (CeCl_3_, 99.9%) were purchased from Macklin. Carbon papers (CPs) were obtained from Shanghai Hesen Electric Co., Ltd., Shanghai, China; 20 wt.% Pt/C was from Aladdin. Concentrated sulfuric acid (H_2_SO_4_, 95.0~98.0%) and nitric acid (HNO_3_, 65.0~68.0%) were obtained from the Guangzhou chemical reagent factory. Nafion D520 (5.0% in isopropanol) was purchased from DuPont. The commercial iridium (IV) oxide (c-IrO_2_, metal basis, Ir ≥ 84.5%) was from Energy Chemical. All the chemicals were used as obtained.

### 2.2. Synthesis of xCeO_2_-Ir/CNTs and CeO_2_/CNTs

CeO_2_-Ir/CNTs (xCeO_2_-Ir/CNTs, x = 1) were synthesized via a solvothermal method. Specifically, 4.0 mL of 5.0 mM IrCl_3_, 4.0 mL of 5.0 mM CeCl_3_, and 3.0 mL of 5.0 g/mL acid-treated CNTs [30] were sequentially added dropwise to 15.0 mL ethanol. Water was added to regulate the volume of the above solution to 30.0 mL, followed by thorough agitation. The obtained mixture was transferred to a 50.0 mL Teflon autoclave and heated at 180.0 °C for 2.0 h. After cooling, the black precipitate was collected by centrifugation and alternate washing with ethanol and water for three runs, followed by vacuum drying. The xCeO_2_-Ir/CNTs were synthesized by varying the feeding volume of CeCl_3_, where x refers to the molar ratio of CeCl_3_/IrCl_3_, albeit keeping other parameters constant. Similarly, CeO_2_/CNTs were synthesized under the same conditions but without IrCl_3_.

### 2.3. Characterization

Morphologies and microstructures of samples were characterized using field-emission scanning electron microscopy (FESEM, Hitachi SU8010) and transmission electron microscopy (TEM, Talos F200X equipped with an energy dispersive X-ray energy spectrometer (EDS). An X-ray diffractometer (XRD, Bruker D8-Advance) with Cu Kα radiation was applied to collect XRD patterns in a scanning range of 10.0 to 90.0° at a rate of 10.0°/min. X-ray photoelectron spectra (XPS) was recorded with an X-ray photoelectron spectrometer (XPS, Thermo ESCA-LAB 250XI) with monochromatic Al Kα excitation source, in which the binding energies were calibrated with C 1 s at 284.6 eV. The elemental composition analysis was conducted on an inductively coupled plasma optical emission spectrometer (ICP-OES, iCAP 7200 Duo). The thermogravimetric (TGA) curve was recorded on a NETZSCH STA 449 F5 under the air stream, starting at 30.0 °C and ramping up to 800.0 °C at a rate of 10.0 °C/min.

### 2.4. Electrochemical Measurements

All electrochemical tests were conducted using a three-electrode system with an electrochemical workstation (CHI 660e, Chenghua, Shanghai). A Ag/AgCl electrode and a Pt wire were utilized as the reference and counter electrodes, respectively. The electrolyte used was a 0.5 M H_2_SO_4_ solution with a pH of 0.32 (Appendix A) at room temperature, consistent with values reported in the existing literature [31,32]. The catalyst-loaded glassy carbon electrode (GCE) was prepared by drop-casting the catalyst ink onto the GCE, and utilized as the working electrode after natural drying. The catalyst ink was prepared by dispersing 4.0 mg catalyst in 1.0 mL of isopropanol/water solution (V/V = 7/3) with 10.0 μL Nafion. The mass loading of the electrocatalysts was estimated to be ~0.20 mg cm^−2^. The linear sweep voltammograms (LSVs) were recorded with a scan rate of 5.0 mV s^−1^ at 1600 rpm to evaluate the catalytic performance of the catalysts. The LSVs reported in this work were 95% iR corrected [33,34,35] unless otherwise stated, and the potentials reported were calibrated with the reversible hydrogen electrode (RHE) using the equation *E*(*V* vs. *RHE*) *= E*(*Ag/AgCl*) *+* 0.059 × *pH*. The electrochemical impedance spectra (EIS) were obtained at 1.32 V in the frequency range from 100 KHz to 0.01 Hz. The cyclic voltammograms (CVs) were measured within the non-Faradic potential range at different scan rates (*v*). The electrochemical double layer capacitance (C_dl_) of catalysts was calculated by plotting half the difference between the anode and cathode current densities (Δ*j = j_anode_ − j_cathode_*) vs. *v*. The electrochemically active surface areas (ECSAs) of the catalysts were caculated according to *S_ECSA_ = C_dl_* × *S/C_s_* (where *S* and *C_s_* represent the geometric area of the electrode and the roughness factor with a value of 35.0 µF cm^−2^ [36], respectively). According to *j_ECSA_ = j_geo_/S_ECSA_*, ECSA-normalized LSVs of the catalysts were estimated (*j_geo_* denotes the geometric activity of the catalysts). The chronopotentiometric curves were used to evaluate the catalysts‘ stability on the carbon paper (CP) with a mass loading of 2.0 mg cm^−2^. The catalyst-loaded CP mentioned in the text (active area: 1.0 × 1.0 cm^2^, Appendix A) was prepared using the same procedure as for GCE.

The Faradaic efficiency (FE) of CeO_2_-Ir/CNTs towards OER on CP was performed in a three-electrode system and investigated using a gas chromatograph (GC, 9560, Shanghai Huaai Scientific Instrument, Shanghai, China). Electrolysis was carried out for a specific duration at a current density of 10.0 mA cm^−2^. Following each electrolysis stage, gaseous samples were drawn from the headspace using a gas-tight syringe (with multiple extractions for averaging), and subsequently analyzed by a GC. Herein, FE of O_2_ was calculated according to equation *FE*(*O*_2_) = *V*(*O*_2_) × 4 × *F*/(*V_m_* × *i* × *t*) × 100%. *V*(*O*_2_) is the generated volume of O_2_, *F* is the Faraday constant of 96,485.3 C/mol, *V_m_* denotes the molar volume of the gas, *i* is the recorded current, and *t* is the time spent in electrolysis.

### 2.5. Evaluation of Water Electrolyzers

Practical usability of the catalyst was appraised by assembling a water electrolyzer using Pt/C and the CeO_2_-Ir/CNTs as the cathode and anode catalysts, respectively. Both Pt/C and CeO_2_-Ir/CNTs were loaded on the CPs. Their mass loadings were both controlled to be 2.0 mg cm^−2^. The electrochemical tests were carried out in 0.5 M H_2_SO_4_. To give a comparison, water electrolyzers with Pt/C and c-IrO_2_ were also prepared.

## 3. Results and Discussion

The synthetic scheme for CeO_2_-Ir/CNTs is presented in Figure 1a. Briefly, CeO_2_-Ir/CNTs were synthesized by maintaining a solution containing CNTs, IrCl_3_, and CeCl_3_ at 180.0 °C for 2.0 h. FESEM was carried out to describe the morphology of CeO_2_-Ir/CNTs, as depicted in Appendix A. The surface of each CNT appeared rough and was densely attached with ultra-fine NPs. The morphologies of Ir/CNTs and CeO_2_/CNTs were similar to that of CeO_2_-Ir/CNTs (Appendix A), indicating that the addition of CeCl_3_ or IrCl_3_ had negligible effects on the samples’ morphologies. The micro-structure of CeO_2_-Ir/CNTs was further investigated by TEM. As depicted in Figure 1b, CeO_2_ and Ir NPs were well-dispersed on the surface of CNT. Size distribution analysis revealed their average diameter was approximately 2.15 nm (Figure 1b). HRTEM, in Figure 1c, demonstrated multiple paired nanoparticles (marked with yellow calabash), where each nanoparticle displayed clear lattice fringes, indicating their excellent crystallinities. The FFT images in Figure 1d,e clearly illustrate that the paired nanoparticles showed different lattice fringe spacings, that is, 0.221 and 0.192 nm, corresponding well to the (111) plane of Ir (PDF No. 87-0715) and the (220) plane of CeO_2_ (PDF No. 89-8436), respectively. These results suggest the successful construction of CeO_2_-Ir heterojunctions (Figure 1c). Furthermore, energy-dispersive X-ray spectroscopy (EDS) mapping demonstrated uniform distribution along the contour of CeO_2_-Ir/CNTs for C, O, Ir, and Ce elements (Figure 1f). XRD patterns of CeO_2_-Ir/CNTs and CeO_2_/CNTs are provided in Figure 2a and Appendix A, respectively. The diffraction peaks of Ir (111), CeO_2_ (111), and (200) of CNTs were observed for CeO_2_-Ir/CNTs, fully consistent with HRTEM observations. Meanwhile, only those of CeO_2_ and CNTs were visible for CeO_2_/CNTs. The composition of xCeO_2_-Ir/CNTs was investigated by ICP-OES and the results are summarized in Appendix A. It was clearly seen that the Ce/Ir molar ratio of xCeO_2_-Ir/CNTs samples was almost consistent with the feeding ratio of CeCl_3_/IrCl_3_. The additional amount of Ce did not alter the structure of xCeO_2_-Ir/CNTs. As depicted in Appendix A, irrespective of the CeO_2_ content in xCeO_2_-Ir/CNTs, their XRD diffraction peaks consistently exhibited a combination pattern comprising CeO_2_, Ir, and CNTs; only the peak of CeO_2_ (111) grew higher with the increasing CeO_2_ content. Thermogravimetric analysis (TGA) was further applied to calculate the weight percentage of the components within CeO_2_-Ir/CNTs. According to Figure 2b, a weight loss of ~88.5 wt.% was obviously observed at 800.0 °C in air, where the CeO_2_-Ir/CNTs was transformed into CeO_2_ and IrO_2_. Consequently, by integrating the ICP-OES and TGA analysis, it was inferred that the weight percentage of Ir in CeO_2_-Ir/CNTs was estimated to be 5.6 wt.%. EDS analysis further confirmed a 1:1 molar ratio of Ce/Ir and an Ir content of 5.6 wt.% in CeO_2_-Ir/CNTs, as shown in Appendix A.

XPS measurement was used to reveal the valance state of CeO_2_-Ir/CNTs. As illustrated in Figure 2c, the survey XPS spectra confirmed the presence of Ce, Ir, O, and C in the sample of CeO_2_-Ir/CNTs. The Ir 4f spectra were deconvoluted into the following two sets of peaks: metallic Ir peaks and oxidized Ir peaks. The peaks at 61.5 and 64.5 eV could be assigned to Ir 4f^7/2^ and Ir 4f^5/2^ of metallic Ir; the oxidized Ir were at 62.9 and 65.9 eV (Figure 2d) [11,37]. The XPS spectra of Ce 3d of CeO_2_-Ir/CNTs are shown in Figure 2e. From Figure 2e, the Ce 3d spectrum could be decomposed into four pairs of spin orbitals (V/U, V′/U′, V″/U″, and V‴/U‴), where V and U represent the 3d^3/2^ and 3d^5/2^ states, respectively [38]. The peaks corresponding to V/U, V″/U″, and V‴/U‴ were attributed to Ce^4+^ 3d orbitals, while the peaks corresponding to V’/U’ were associated with Ce^3+^ 3d [39]. The ratio of Ce^3+^/Ce^3+^ + Ce^4+^ in CeO_2_ kept constant (39%) after coupling with Ir (Figure 2e), indicating that the chemical environment of Ce did not undergo significant changes. The O 1s spectra in Figure 2f could be deconvoluted into three peaks: O_L_ (531.0 eV), Vo· (532.0 eV), and O-H (533.6 eV), where O_L_ and Vo· refer to oxygen in the lattice and oxygen vacancies, respectively. The Vo· peak further verified the existence of Vo· in the CeO_2_, in line with the presence of a large amount of Ce^3+^ species. Closer examinations revealed a strong electron coupling between CeO_2_ and Ir through the heterointerface in the CeO_2_-Ir/CNTs. Figure 2d shows that the peaks corresponding to Ir 4f in the CeO_2_-Ir/CNTs appeared at a relatively higher binding energy compared with Ir/CNTs (synthesized in the absence of Ce). Remarkably, after being coupled with Ir, the O 1s peaks of CeO_2_-Ir/CNTs appeared at a relatively lower binding energy compared with CeO_2_/CNTs (synthesized in the absence of Ir). These observations support the conclusion that the coupling between CeO_2_ and Ir NPs results in strong electron coupling, particularly occurring between Ir and the O_L_ of CeO_2_.

The influence of CeO_2_ content on the catalytic activity of xCeO_2_-Ir/CNTs was investigated. As illustrated in Appendix A, the catalyst activity was improved, by increasing x from 0 to 1.0. However, further increasing x to 1.5 led to declined activity. These findings suggest that optimal OER activity could be achieved by maintaining the Ce/Ir molar ratio of xCeO_2_-Ir/CNTs at 1/1. Furthermore, Tafel plots (Appendix A) and EIS analysis (Appendix A) revealed that the OER kinetics and charge transfer resistance of xCeO_2_-Ir/CNTs highly depended on x, and both reached the best at x = 1.0. The electrochemical performance of the synthesized CeO_2_/CNTs and the purchased c-IrO_2_ were also studied as comparisons. Specifically, CeO_2_-Ir/CNTs demonstrated remarkable catalytic activity in the acidic OER test. As depicted in Figure 3a, CeO_2_-Ir/CNTs demonstrated remarkable catalytic activity with an overpotential of 262.9 mV at 10.0 mA cm^−2^ under the acidic OER conditions (0.5 M H_2_SO_4_), surpassing that of Ir/CNTs (285.9 mV), c-IrO_2_ (304.1 mV), and CeO_2_/CNTs (with negligible OER current). Furthermore, CeO_2_-Ir/CNTs even outperformed numerous recently reported Ir-based catalysts (Appendix A). The superior catalytic activity of CeO_2_-Ir/CNTs was further evidenced by its accelerated OER kinetics. As illustrated in Figure 3b, CeO_2_-Ir/CNTs exhibited a Tafel slope of 53.4 mV dec^−1^, lower than that of Ir/CNTs (59.2 mV dec^−1^) and c-IrO_2_ (69.5 mV dec^−1^). Additionally, mass activities of CeO_2_-Ir/CNTs, Ir/CNTs, and c-IrO_2_ are plotted in Figure 3c. Among them, CeO_2_-Ir/CNTs exhibited a mass activity of 2542.3 A g_Ir_^−1^ at 1.53 V, 1.8 and 58.8 times higher than that of Ir/CNTs and c-IrO_2_, respectively. EIS was performed to elucidate the origin of the superior catalytic activity of CeO_2_-Ir/CNTs. At a potential of 1.32 V, CeO_2_-Ir/CNTs displayed a significantly reduced charge transfer resistance value of 18.1 Ω compared with c-IrO_2_ (46.5 Ω) and other catalysts (Figure 3d). CeO_2_-Ir/CNTs efficiently converted electrical power into chemical power, as evidenced by its remarkable FE of 98.2% (Appendix A). It is important to note that the FE of CeO_2_-Ir/CNTs was less than 100%, which could be attributed to minor carbon corrosion during the OER [40].

CV tests were conducted at different scan rates (*v*) in the non-Faradic region to calculate the C_dl_ of catalysts, as depicted in Appendix A. To exclude the morphological effects caused by CeO_2_ and to investigate the intrinsic catalytic activity of the catalysts, LSVs for xCeO_2_-Ir/CNTs were normalized by their ECSAs that were determined by using a CV method, as depicted in Appendix A. Notably, CeO_2_-Ir/CNTs displayed a relatively larger ECSA (46.0 cm^2^) compared with Ir/CNTs (42.1 cm^2^), 0.5CeO_2_-Ir/CNTs (45.4 cm^2^), and 1.5CeO_2_-Ir/CNTs (41.5 cm^2^). Nevertheless, after the ECSA normalization, CeO_2_-Ir/CNTs still showed a larger current density than others, indicating their superior intrinsic activity (Appendix A). Thus, the CeO_2_ content had a notable influence on the intrinsic catalytic activity of xCeO_2_-Ir/CNTs. Considering the XPS analysis presented in Figure 2d,f, it could be inferred that the electron interaction between O_L_ of CeO_2_ and Ir led to the generation of electron-deficient Ir atoms. DFT calculations in a recently reported paper [41] similarly showed that, when Ir NPs interacted with CeO_2_ NPs, Ir-O bonds formed at the interfaces, allowing for electron transfer from Ir atoms to the O_L_ of CeO_2_. The charge density of Ir atoms was adjusted by forming Ir-O bonds between Ir and CeO_2_ in order to optimize the adsorption/desorption of intermediates and finally improve the catalytic activity of OER. This phenomenon could be recognized as the primary factor responsible for the observed enhancement in the intrinsic catalytic activity of CeO_2_-Ir/CNTs. The durability of OER catalysts is a crucial factor that must be carefully assessed to determine their practicality. To further investigate the stability of CeO_2_-Ir/CNTs under the acidic OER conditions, we conducted chronopotentiometry at 10.0 mA cm^−2^. Figure 3e illustrates that, even after prolonged testing exceeding more than 60.0 h, CeO_2_-lr/CNTs exhibited excellent durability in the acidic OER test with no significant degradation of their catalytic activity. In contrast, c-IrO_2_ and Ir/CNTs were completely deactivated within less than 21.0 h and 31.0 h, respectively.

To confirm the high durability of CeO_2_-Ir/CNTs in acidic OER tests, the CeO_2_-Ir/CNTs’ sample after the stability test was characterized using TEM, HRTEM, and XPS. Firstly, TEM analysis confirmed that the morphology of CeO_2_-Ir/CNTs remained intact with exceptional structural robustness (Figure 4a). The distribution of CeO_2_ and Ir NPs on CNTs could remain uniform (Figure 4b). The paired nanoparticles in Figure 4b retained good crystallinity with lattice fringe spacings of 0.221 nm and 0.192 nm, corresponding to Ir (111) and CeO_2_ (220), respectively (Figure 4c,d). Additionally, XPS characterization was performed on CeO_2_-Ir/CNTs to investigate if there was compositional decay during the stability test. Figure 4e–g show that the CeO_2_-Ir/CNTs exhibited the XPS spectra of Ir 4f, Ce 3d, and O1s, with the profiles comparable to those before the OER; however, the weaker Ce 3d profile (Figure 4f) may have been attributed to the dissolution of CeO_2_. The component analysis revealed a slight decrease in the Ir content of CeO_2_-Ir/CNTs after stability testing, while the loss of CeO_2_ was more pronounced (Appendix A). Therefore, it can be concluded that excellent structural integrity along with stable component composition contributes significantly to maintaining superior activity during acidic OER.

To further validate the potential application of CeO_2_-Ir/CNTs in acidic water electrolyzers, it was evaluated as an anode catalyst for over-all water splitting (OWS). Figure 5a demonstrates the assembly and testing of Pt/C||CeO_2_-Ir/CNTs as the electrode pair in the acidic water electrolyzers. LSVs and chronopotentiometric curves were employed to estimate their OWS performance. Remarkably, a current density of 10.0 mA cm^−2^ could be achieved with an input voltage of only 1.54 V for Pt/C||CeO_2_-Ir/CNTs, while the commercial electrode pair of Pt/C||c-IrO_2_ required a larger input voltage of 1.58 V to achieve the same current density (Figure 5b). The other synthesized samples were assembled with Pt/C to construct electrolyzers and evaluate their performance towards OWS, as depicted in Appendix A. Furthermore, Figure 5c illustrates that Pt/C||CeO_2_-Ir/CNTs could maintain continuous electrolysis for over 60.0 h without significant loss of activity at 10.0 mA cm^−2^; however, Pt/C||c-IrO_2_ became completely inactive after less than 20.0 h under similar conditions. These results strongly demonstrate the promising potential of CeO_2_-Ir/CNTs as efficient anode catalysts for acidic water electrolyzers.

## 4. Conclusions

We successfully demonstrated the synthesis of a series of xCeO_2_-Ir/CNTs with varying CeO_2_ contents using a facile solvothermal method. The catalytic activity of xCeO_2_-Ir/CNTs was significantly influenced by the content of CeO_2_, with higher catalytic activity observed for acidic OER when the molar ratio of CeO_2_/Ir was 1:1. Due to the strong electron coupling in CeO_2_-Ir heterojunctions, CeO_2_-Ir/CNTs exhibited outstanding OER catalytic performance in acidic media, significantly exceeding the benchmark c-IrO_2_. CNTs played a crucial role by providing abundant nucleation sites and excellent conductive pathways for CeO_2_-Ir, contributing to the observed high catalytic and mass activity in CeO_2_-Ir/CNTs. The acidic water electrolyzers constructed using CeO_2_-Ir/CNTs in combination with Pt/C (Pt/C||CeO_2_-Ir/CNTs) demonstrated significant potential for practical applications and outperformed the benchmark c-IrO_2_||Pt/C system. This study presents a novel approach for the design of efficient, stable, and low-Ir-usage catalysts towards acidic OER.

## Figures and Tables

**Figure 1 materials-16-07000-f001:**
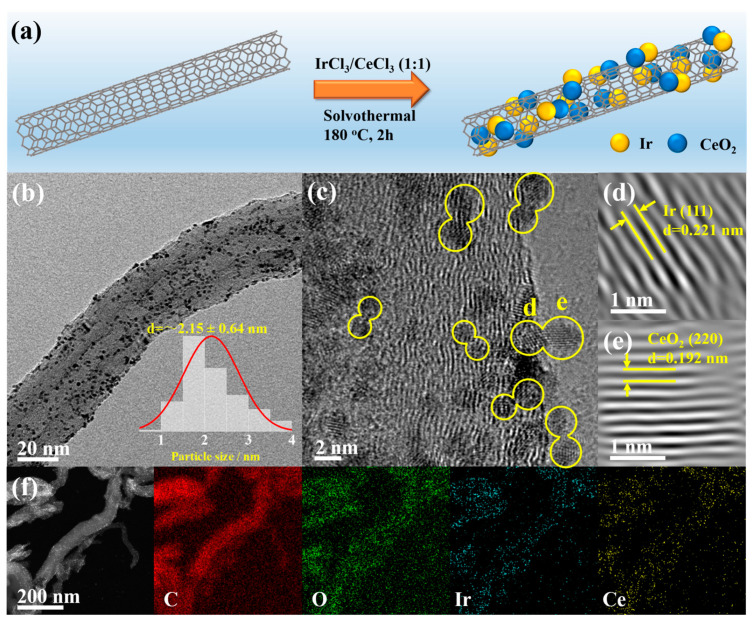
(**a**) Brief illustration of the synthetic procedure for CeO_2_-Ir/CNTs. (**b**) TEM, (**c**) HRTEM, and (**f**) EDS mapping images of CeO_2_-Ir/CNTs. (**d**,**e**) represent Ir and CeO_2_ nanoparticle lattice fringes from (**c**), respectively.

**Figure 2 materials-16-07000-f002:**
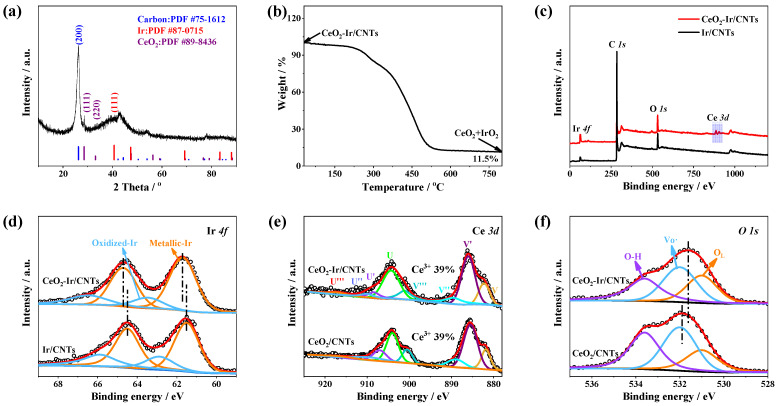
(**a**) XRD pattern and (**b**) TGA curve of CeO_2_-Ir/CNTs. (**c**) Survey XPS spectra. XPS spectra of (**d**) Ir 4f, (**e**) Ce 3d, (**f**) O1s.

**Figure 3 materials-16-07000-f003:**
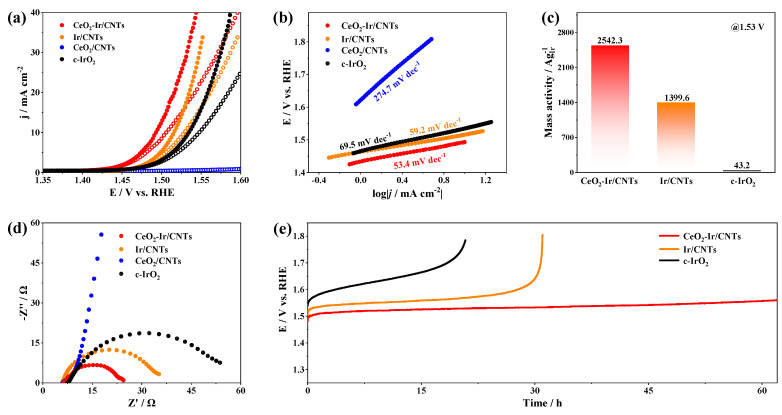
(**a**) LSVs (solid−dot line with iR compensation), (**b**) Tafel slopes, (**c**) mass activity, (**d**) EIS Nyquist plots, and (**e**) chronopotentiometric curves of different catalysts.

**Figure 4 materials-16-07000-f004:**
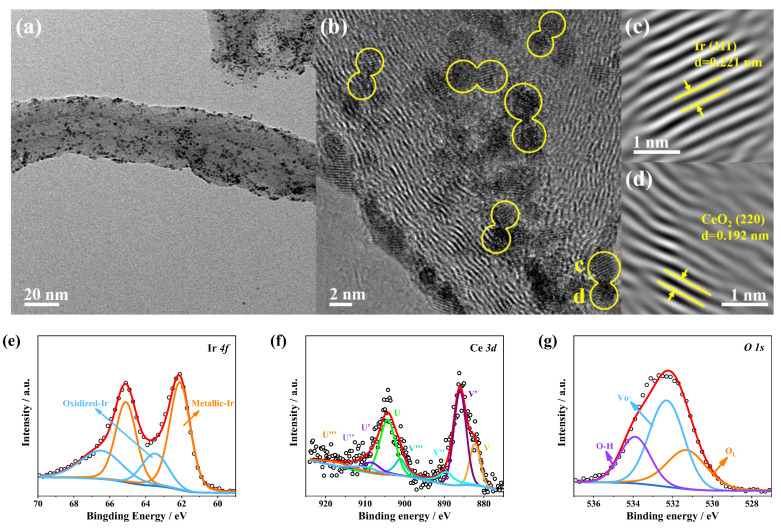
(**a**) TEM, (**b**) HRTEM images, and (**e**–**g**) XPS spectra of CeO_2_-Ir/CNTs after OER stability test. (**c**,**d**) represent Ir and CeO_2_ nanoparticle lattice fringes from (**b**), respectively.

**Figure 5 materials-16-07000-f005:**
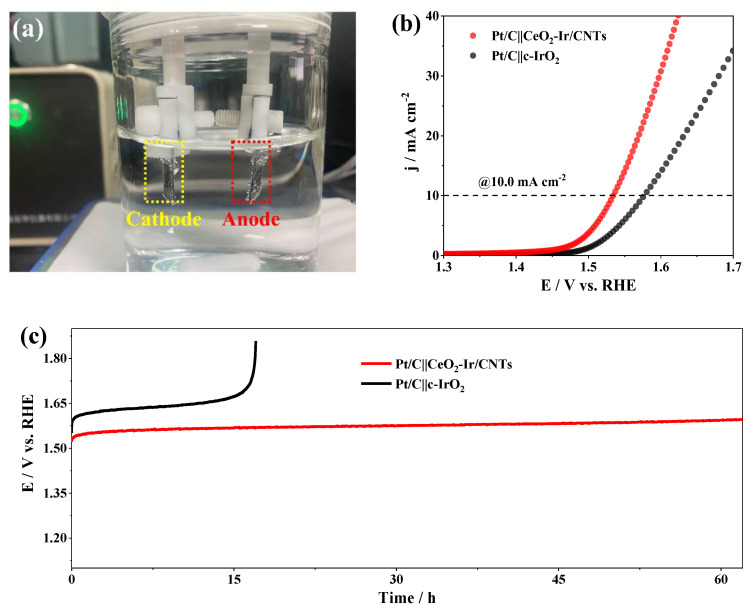
(**a**) Illustration of acidic water electrolyzers. (**b**) LSVs and (**c**) chronopotentiometric curves of Pt/C||CeO_2_-Ir/CNTs and Pt/C||c-IrO_2_ in water electrolyzers using 0.5 M H_2_SO_4_ as electrolyte.

## Data Availability

The authors are able to provide the data presented in this study upon request.

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
