# Peer review of "Achieving High Activity and Long-Term Stability towards Oxygen Evolution in Acid by Phase Coupling between CeO2-Ir"

_materials, 2023, doi:10.3390/ma16217000_

Round 1

Reviewer 1 Report

Comments and Suggestions for Authors

Manuscript focus on search of effective catalysts  for acidic oxygen evolution reaction (OER). Authors prepare and characterized by ICP, XRD, XPS,TEM-EDX and TGA series of Ir doped CeO2 materials supported on carbon nanotubes. After deposition on CPs selected materials were tested in OER. The results confirm significantly higher catalytic performance CeO2-Ir/CNTs in compression with commercial c-IrO2. In general, the results appear some interest for readers. I recommend publication of the manuscript after major revisions suggested below.

1. In introduction (last paragraph) novelty have to be described. Moreover, the goal of the manuscript cannot be the recipe for catalysts preparation or conclusions. The last paragraph in the introduction should be reconstructed.

 2. Notation of xCeO2-Ir/CNTs and CeO2-Ir/CNTs catalysts should be unified. Do the CeO2-Ir/CNTs and 1CeO2-Ir/CNTs (where x=1) names mean the same catalyst?

3. In the Materials and Methods section, some additional information should be added. These information are needed for reconstruction in others laboratories. For example: TGA temperature rate,  XRD  diffractometer operated conditions and step size, etc. Moreover, in 2.5 paragraph the way of catalysts loading on the CPs should be explained.

4. CeO2-Ir/CNTs was evaluated as an anode catalyst for over-all water splitting (OWS). What about others prepared xCeO2-Ir/CNTs materials performance? Author should also reports the results for Ir/CNTs and CeO2/CNTs as anode catalysts for acidic water electrolyzers.

5. Authors suggest (base of Table s3) that after acid OER tests content of Ir remained constant while those of CeO2 slightly decayed. But the ICP results show rather significant loss of Ce in the sample after OER (about 25 %). Please explained reason of Ce loss? Did cerium was washed out from the electrode?

Author Response

Thank you for your valuable comments, and we have carefully revised the manuscript according to your suggestions.

Reviewer 2 Report

Comments and Suggestions for Authors

The manuscript under the title "Achieving High Activity and Long-Term Stability towards Oxygen Evolution in Acid by Phases Coupling between CeO2-Ir" stands as a noteworthy contribution to the field. This study synthesizes xCeO2-Ir/CNTs catalysts with variable CeO2 content through solvothermal methods. CeO2-Ir/CNTs exhibit superior catalytic performance in the acidic Oxygen Evolution Reaction (OER), outperforming c-IrO2 and other catalysts. Carbon Nanotubes enhance nucleation and conductivity. Despite lower Ir dosage, CeO2-Ir/CNTs achieve significantly higher mass activity, offering a novel, efficient, and low-iridium solution for acidic OER applications.

The introduction adeptly provides a comprehensive background, ensuring that readers grasp the contextual relevance and significance of the research. The methodology section is suitably elaborated, offering lucidity regarding the experimental methodology employed. Data interpretation and discussion are articulated coherently and meaningfully. The manuscript exhibits a well-structured narrative with a logical flow throughout. Moreover, the references are pertinent, and seamlessly integrated into the text. In its current form, the manuscript appears to align with the requisite standards for publication in the Materials.

Author Response

Thanks to the evaluation.

Reviewer 3 Report

Comments and Suggestions for Authors

            In this present work, the author reported a “Achieving High Activity and Long-Term Stability towards Oxygen Evolution in Acid by Phases Coupling between CeO2-Ir works are interesting. Moreover, the CeO2-Ir heterojunctions supported on carbon nanotubes (CeO2-Ir/CNTs) are synthesized using a solvothermal method based on the heterostructure strategy. CeO2-Ir/CNTs demonstrate remarkable effectiveness as catalysts for acidic OER, achieving 10 mA cm-2 at a low overpotential of only 262.9 mV and maintaining stability over 60 h. Notably, despite using an Ir dosage 15.3 times lower than that of c-IrO2, CeO2-Ir/CNTs exhibit a very high mass activity (2542.3 A gIr-1@1.53 V), which is 58.8 times higher than that of c-IrO2. When applied to acidic water electrolyzes, CeO2-Ir/CNTs display a prosperous potential for application as an anodic catalyst. The overall scientific organization of this article is satisfactory with some minor revisions as must be needed before acceptance. I have the following concerns.

1.      What is the benefit of the selected composition in CeO2-Ir/CNTs? Can you check with changes in the ratio of Ce and Ir concentration?

2.      Carbon is not stable under high working potential during OER. The authors should be careful when using carbon as an OER catalyst substrate (e.g., Angew. Chem. Int. Ed. 2020, 59, 1585-1589). Moreover, there may be no apparent change in the electrolyte colour during carbon corrosion as carbon will be oxidized into carbon dioxide. Hence, the electrolyte should be checked for carbon corrosion and presented with valid data.

3.      The authors must provide quantitative data to prove the composition. Could you explain the weight % of as-synthesized CeO2-Ir/CNTs catalysts by either EDS and/or XPS spectra?

4.      Consistently express the units used in the manuscript. Please keep the Figure. * or Fig.* style consistent throughout the main content. Also, use either ml or mL as an example.

5.      Understanding the structural transformation mechanism and the active species is an essential precondition for designing highly efficient OER catalysts.

6.      Abstract and conclusion part since it's more identical. Authors should revise with the different also the proper key results including numerical hints.

Comments on the Quality of English Language

Minor editing of English language required

Author Response

(The authors gave the same response as above.)

Reviewer 4 Report

Comments and Suggestions for Authors

Using a solvothermal method and a heterostructure strategy, Kuang et al. synthesized CeO2-Ir heterojunctions supported on carbon nanotubes (CeO2-Ir/CNTs). These materials exhibited exceptional catalytic performance in the acidic OER, achieving a current density of 10 mA cm-2 with a low overpotential of 262.9 mV and maintaining stability for 60 hours. Impressively, despite using only 6.5% of the Ir dosage compared to conventional c-IrO2, CeO2-Ir/CNTs demonstrated a much higher mass activity (2542.3 A gIr-1@1.53 V), surpassing c-IrO2 by a factor of 58.8. Additionally, when tested in acidic water electrolysis, CeO2-Ir/CNTs showed promise as an anodic catalyst.

Comments:

1. Authors can show a synthetic scheme for synthesis of xCeO2-Ir/CNTs and CeO2/CNTs.

2. “Considering the XPS analysis presented in both Figure 2d and Figure 2f, it can be inferred that the electron interaction between CeO2 and Ir leads to the generation of electron-deficient Ir. This phenomenon is recognized as the primary factor responsible for the observed enhancement in the intrinsic catalytic activity of CeO2-Ir/CNTs[15, 33].” Authors should explain the electron interaction mechanism between CeO2 and Ir and how it creates electron-deficient Ir. Authors should also comment on the change in electrochemical band gap of CeO2/CNTs, Ir/CNTs, and CeO2-Ir/CNTs.

Author Response

(The authors gave the same response as above.)

Reviewer 5 Report

Comments and Suggestions for Authors

This work reports the synthesis of CeO2-Ir heterostructures supported on carbon nanotubes (CeO2-Ir/CNTs) via a solvothermal route demonstrating an effective catalyst for acidic OER with at an overpotential of 262.9 mV at 10 mA cm-2 and maintaining the catalytic activity over 60 h. The work looks interesting and novel. However, the manuscript needs to be revised and further evaluated on the following points.

(1) Introduction part should be improved by discussing the significance of heterojunction structures and application of Ce-based electrocatalysts in OER. For example, please see the recent work: Mater. Today Phys. 2023, 38, 101252.

(2)   SEM analysis of the CeO2-Ir/CNTs and CeO2/CNTs should be performed.

(3)   The XRD patterns of the CeO2/CNTs shown in Fig. S1 and that of CeO2-Ir/CNTs shown in Fig. 2a do not match with the reference CeO2:PDF#81-0792.

(4)  What is the active material for OER in this work? What was the atomic ratio of Ce and Ir in CeO2-Ir/CNTs?

(5)  A detailed mechanism for the enhanced OER activity by the CeO2-Ir/CNTs should be discussed.

(6) What was the % of iR loss compensation during the measurement of LSV polarization curves? The series resistance Rs in Fig. S2 and Fig. 3d looks large. Therefore, the authors should also add the LVS curves without iR loss compensation.

(7)   What was the actual measured pH of the electrolyte?

Comments on the Quality of English Language

English of this manuscript looks fine. 

Author Response

(The authors gave the same response as above.)

Round 2

Reviewer 1 Report

Comments and Suggestions for Authors

any

Reviewer 5 Report

Comments and Suggestions for Authors

The authors have revised the manuscript carefully. The revised manuscript is suitable to be accepted.